# Oxidation-specific epitopes restrain bone formation

Elena Ambrogini [1], Xuchu Que[2], Shuling Wang[2], Fumihiro Yamaguchi [2], Robert S. Weinstein[1], Sotirios Tsimikas[3], Stavros C. Manolagas[1], Joseph L. Witztum[2] & Robert L. Jilka[1]

Atherosclerosis and osteoporosis are epidemiologically linked and oxidation specific epitopes (OSEs), such as phosphocholine (PC) of oxidized phospholipids (PC-OxPL) and malondialdehyde (MDA), are pathogenic in both. The proatherogenic effects of OSEs are opposed by innate immune antibodies. Here we show that high-fat diet (HFD)-induced bone loss is attenuated in mice expressing a single chain variable region fragment of the IgM E06 (E06-scFv) that neutralizes PC-OxPL, by increasing osteoblast number and stimulating bone formation. Similarly, HFD-induced bone loss is attenuated in mice expressing IK17-scFv, which neutralizes MDA. Notably, E06-scFv also increases bone mass in mice fed a normal diet. Moreover, the levels of anti-PC IgM decrease in aged mice. We conclude that OSEs, whether produced chronically or increased by HFD, restrain bone formation, and that diminished defense against OSEs may contribute to age-related bone loss. Anti-OSEs, therefore, may represent a novel therapeutic approach against osteoporosis and atherosclerosis simultaneously.

---

[1] Division of Endocrinology and Metabolism, Center for Osteoporosis and Metabolic Bone Diseases, University of Arkansas for Medical Sciences and Central Arkansas Veterans Healthcare System, 4301W. Markham, Little Rock, AR 72205, USA. [2] Division of Endocrinology and Metabolism, University of California San Diego, La Jolla, CA 92093-0682, USA. [3] Department of Medicine, Cardiololgy, University of California San Diego, 9500 GilmanDrive, La Jolla, CA 92093-0682, USA. Correspondence and requests for materials should be addressed to E.A. (email: eambrogini@uams.edu)

Epidemiologic studies have established a strong correlation between atherosclerosis and osteoporosis, suggesting that a common pathogenic mechanism may underlie both conditions[1–4]. In a meta-analysis of twenty-five studies involving more than 10,000 subjects, atherosclerotic lesions were significantly increased in patients with osteopenia and osteoporosis independently of age, sex, body mass index, and other cardiovascular risk factors[5]. Consistent with the human data, mouse models of hyperlipidemia/atherosclerosis exhibit decreased bone mass, primarily due to a reduction in osteoblast number and bone formation[6–10].

Oxidation of polyunsaturated fatty acids (PUFAs) by reactive oxygen species (ROS) is a feature of many physiologic and pathologic processes, including apoptosis, cellular senescence, and inflammation[11], as well as the bone loss that accompanies atherosclerosis in rodents and humans[9,10]. Lipid peroxidation generates highly reactive degradation products such as malondialdehyde (MDA), 4-hydroxynonenal (4-HNE), and oxidized phospholipids (OxPL), such as oxidized phosphatidylcholine. These moieties react with amino groups on proteins and other lipids to form adducts that are collectively known as oxidation-specific epitopes (OSEs)[11]. OSEs are members of a larger group of proinflammatory and immunogenic molecules—known collectively as damage-associated molecular patterns (DAMPs)—that are produced by excessive ROS, necrosis, ischemia reperfusion, or chemically induced tissue injury[11–13]. DAMPs often share structural homology with moieties present on microbes, which are collectively known as pathogen-associated molecular patterns (PAMPs). PAMPs and DAMPs bind to evolutionary conserved pattern recognition receptors (PRRs)[14]. Binding of PAMPs to PRRs activates mechanisms that kill invading microbes. Binding of DAMPS to PRRs sequesters them and also mounts defenses that attempt to prevent cell damage.

The PRRs are either cell bound, such as the large family of scavenger receptors (SRs) and toll-like receptors (TLRs), or soluble, such as the natural antibodies (NAbs) produced by B-1 lymphocytes[14–18]. NAbs are predominantly of the IgM class and comprise about 80% of the total serum IgM in uninfected mice[19]. Their antigen binding sites are generated by rearrangement of germline-encoded variable region genes in the complete absence of foreign antigen exposure. NAbs constitute the first line of defense against microbial pathogens. In a similar manner, NAbs may also maintain homeostasis against OSEs, which are ubiquitous and increased in inflammatory states[11]. In addition, because some OSEs share molecular signatures with PAMPs, there may have been selective pressure from both OSEs and PAMPs for the preservation of NAbs that bind to both[11].

The IgM NAb E06 recognizes the phosphocholine (PC) moiety of oxidized phosphatidylcholine present on the membrane of apoptotic cells and oxidized low-density lipoproteins (OxLDLs)[20]. The PC moiety is normally inaccessible to PRRs, but it is exposed following oxidation of the sn2 PUFA of phosphatidylcholine and thereby becomes a conformational neo-epitope (PC-OxPL)[20]. Importantly, E06 prevents the binding of OxLDL by SRs and TLRs on macrophages and thereby reduces the proinflammatory activity of OxPLs[11]. Similar to E06, both the murine NAb E014 and the human IK17, recognize MDA and prevent the proinflammatory effect of MDA on macrophages[11,21].

Emerging evidence indicates that when OSEs persist and/or are excessive, the protective effect of the innate immune system against them is overwhelmed. Under such circumstances, the inflammatory response initiated by TLR and SR activation eventually leads to tissue damage and disease[11]. In atherosclerosis, OSEs in OxLDL bind to their cognate receptors present in endothelial cells and macrophages and stimulate the production of inflammatory cytokines, which in turn recruit and activate additional macrophages and lymphocytes in a pathologic cascade that eventually leads to the development of atheroma, vascular inflammation and calcification[15–17]. Consistent with their anti-OSE properties, administration of either E06 IgM or IK17-scFv—a single chain variant of the antigen binding domain of the human anti-MDA antibody (IK17)—prevents the formation of foam cells and atheroma in murine models of atherosclerosis[15,22–24]. Moreover, genetic maneuvers that increase overall levels of anti-OSE antibodies in mice attenuate atherosclerosis[25,26].

In bone, SRs and TLRs are present on osteoblasts, osteoclasts, and bone marrow macrophages, raising the possibility that physiologic or pathologic generation of OSEs could affect bone homeostasis[27,28]. Previous studies have demonstrated in murine models of atherosclerosis that OxLDL stimulates a pro-inflammatory response of bone and loss of bone mass[10]. Additionally, OxLDL attenuates osteoblast generation and promotes osteoblast apoptosis in vitro[9,29–33]. Because of this evidence, we investigated the role of OSEs in bone homeostasis. To do this, we used mice overexpressing the antigen recognition portion of E06 (E06-scFv), or IK17-scFv, which block the effects of PC-OxPL or MDA, respectively. We found that the E06-scFv transgene increases cancellous bone mass and attenuates the loss of cortical bone mass caused by high fat diet (HFD) in the LDL-receptor knockout (LDLR-KO) model of atherosclerosis. An increase in cancellous, as well as cortical, bone mass is observed also in HFD-fed LDLR-KO mice expressing the IK17-scFv transgene. Unexpectedly, the bone anabolic properties of E06-scFv on cancellous bone is present not only in the setting of enforced atherogenesis, but also in C57BL/6 J mice maintained on a normal diet. Lastly, consistent with evidence in humans that B-1 cells decline with age[34–37] and low levels of anti-PC IgM antibodies are associated with increased incidence of cardiovascular diseases[11], we show that sera from 22- to 26-month-old female C57BL/6 J mice have dramatically decreased levels of anti-PC IgM as compared to 6- or 7-month-old mice. These findings indicate that OSEs, produced chronically and in response to HFD, restrain bone formation in mice; and that the age-related bone loss might be due in part to diminished defense against OSEs. Anti-OSEs, therefore, may represent a novel therapeutic approach to the prevention and treatment of osteoporosis and atherosclerosis simultaneously.

## Results

**E06 prevents the effects of OxLDL on osteoblastic cells.** Consistent with earlier reports[9,29–33], addition of OxLDL to cultured osteoblastic cells attenuated their proliferation and differentiation, and stimulated apoptosis (Fig. 1a–c). All these effects were prevented by the monoclonal murine E06 IgM, but not by IgM obtained from normal mice (Fig. 1a–c). E06 IgM by itself had no effect on any of these parameters in the absence of OxLDL.

We made transgenic C57BL/6 J mice expressing a single-chain variable region fragment (scFv) of the E06 IgM. The transgene (E06-scFv) consists of a fusion protein of the heavy and light-chain variable domains covalently joined by a flexible peptide linker (Fig. 2a). The transgene is expressed under the control of the Apo-E promoter to provide for a high level of secretion of the protein by the liver and macrophages. E06-scFv transgenic mice had normal body weight and did not display any morphologic or reproductive abnormalities[38]. The His-tagged E06-scFv protein was detected by ELISA in the serum of E06-scFv transgenic mice, but not in WT mice (Fig. 2b). The E06-scFv is present in plasma of mice at a concentration of 20–30 µg/ml, and the level of endogenous IgM E06 was not affected by the expression of the

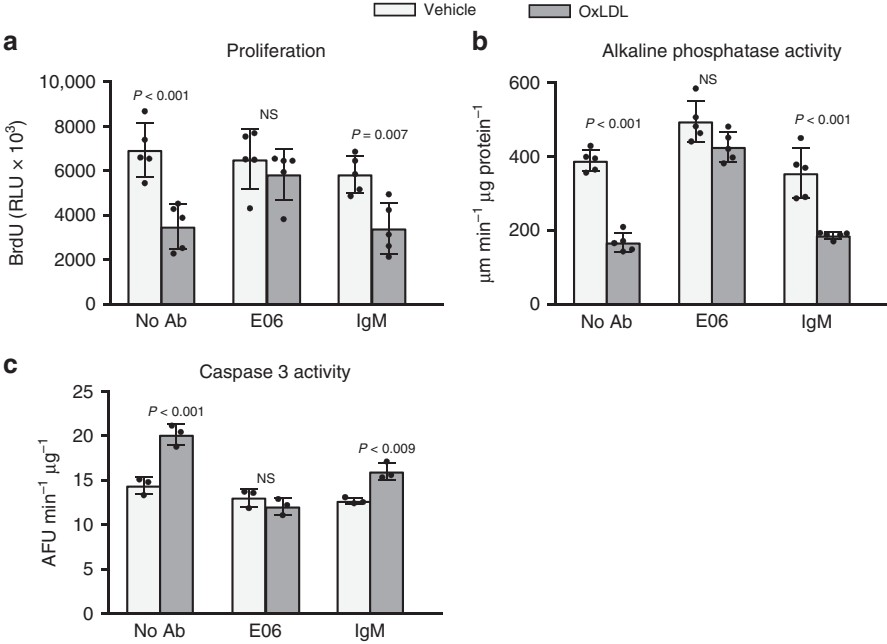

**Fig. 1** E06 prevents the effects of OxLDL on osteoblast apoptosis, differentiation and proliferation. Osteoprogenitor cells from bone marrow were pre-treated for 1 h without (no Ab), or with E06 (2 μg/ml), or with murine IgM ('IgM', 2 μg/ml), in the absence ('veh') or presence of OxLDL (50 μg/ml) for **a** 2 days prior to measurement of BrDU incorporation, or **b** 2 days prior to measurement of alkaline phosphatase activity, or **c** 12 h prior to measurement of Caspase 3 activity. All measures were performed in triplicate cultures. Essentially identical results were obtained in a separate experiment. Data were analyzed by ANOVA; the p values reflect comparison to the respective vehicle. Data shown represent the mean (±SD). AFU: Arbitrary fluorescence units, RLU: Relative luminescence units, NS: not significant

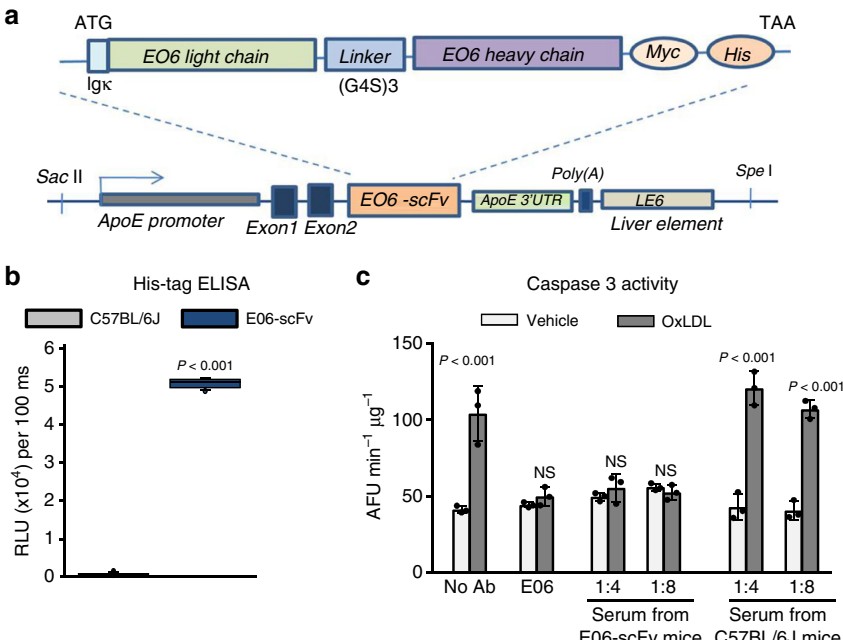

**Fig. 2** Generation of E06-scFv transgenic mice. **a** E06-scFv transgene structure. The transgene consists of a fusion protein of the heavy- and light-chain variable domains, which constitute the antigen recognizing portion of the natural antibody E06. The domains are covalently linked by a flexible peptide linker. The transgene is expressed under the control of the ApoE promoter. Reproduced from Que et al.[38] with permission. **b** Quantification of E06-scFv in the serum of C57BL/6 J and E06-scFv 5-month-old male mice as determined by ELISA. C57BL/6 J n = 5, E06-scFv n = 6. Data were analyzed by Rank Sum test. **c** E06-scFv prevents the pro-apoptotic effect of OxLDL. Osteoblastic cells obtained from neonatal calvaria were pre-treated for 1 h without (no Ab), or with E06 (2 μg/ml), or with serum from E06-scFv or C57BL/6 J mice at the indicated dilutions (1:4, 1:8). Serum from 4 mice (per genotype) was pooled for this analysis. Caspase 3 activity was then measured 6 h after addition of OxLDL (50 μg/ml). All measures were performed in triplicate cultures. Essentially identical results were obtained in a separate experiment. Data were analyzed by ANOVA; the p values reflect comparison to the respective vehicle. Data shown represent the mean (±SD). AFU: Arbitrary fluorescence units

E06-scFv antibody[38]. Similar to the effect of the E06 IgM, serum from E06-scFv transgenic mice prevented OxLDL-induced osteoblast apoptosis while serum from WT mice did not (Fig. 2c). The presence of the transgene did not change plasma lipid or lipoprotein levels in response to the various diets[38].

**E06-scFv attenuates cortical bone loss in HFD-fed mice.** Two-month-old LDLR-KO male mice were placed on a normal diet (ND), or on a HFD containing 0.5% cholesterol and 21% milk fat. A cohort of LDLR-KO;E06-scFv mice was placed on the HFD at the same age. At 6.5 months, the LDLR-KO mice on HFD had lower femoral bone mineral density (BMD) compared to the LDLR-KO mice fed ND, as determined by dual-energy X-ray absorptiometry (DXA) (Fig. 3a). Strikingly, LDLR-KO;E06-scFv mice were protected from the adverse effect of HFD on femoral BMD. At 7.5 months of age, the mice were euthanized and both femoral cortical and cancellous compartments were analyzed by micro-CT and histology. The HFD reduced cortical diaphyseal

thickness (Fig. 3b) in LDLR-KO mice. This effect was due to reduced periosteal apposition, as evidenced by a decrease in the total area without a change in the medullary area. HFD-fed LDLR-KO;E06-scFv mice had similar cortical thickness and cortical bone area as LDLR-KO mice on normal diet (Fig. 3b). However, the E06-ScFv transgene preserved cortical thickness in the HFD-fed LDLR-KO;E06-ScFv mice by decreasing medullary area. Thus, E06-scFv increased endosteal bone apposition, but did not prevent the negative effects of HFD on periosteal expansion (Fig. 3b). Endosteal osteoblast number and bone formation rate were reduced in HFD-fed LDLR-KO mice as compared to the ND controls, but these effects were prevented in LDLR-KO;E06-scFv mice (Fig. 3c). Neither the HFD nor the E06-scFv transgene affected osteoclast number.

**E06-scFv increases cancellous bone in HFD-fed mice.** Femoral metaphyseal cancellous bone mass, measured by micro-CT, was unaffected by the HFD; albeit, there being a modest increase in

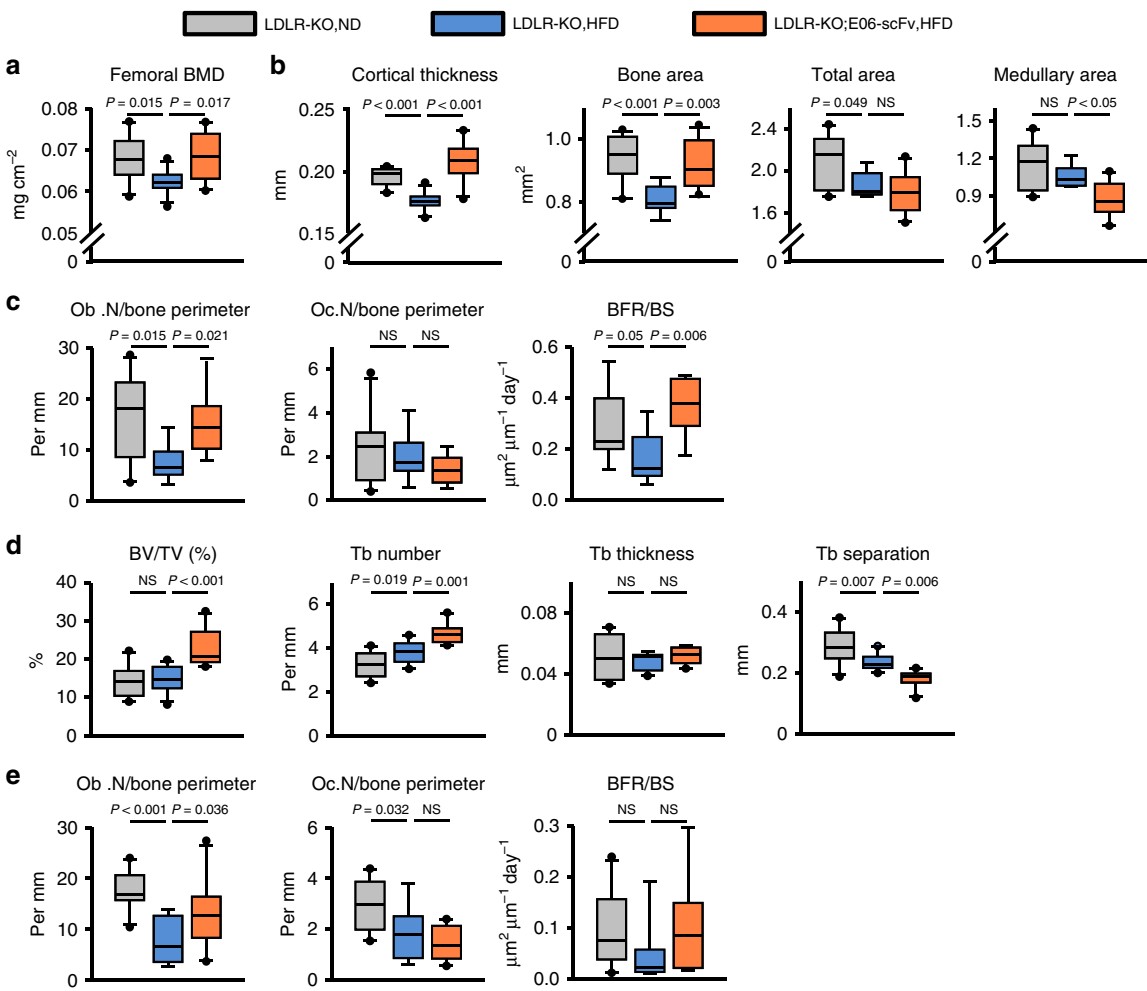

**Fig. 3** E06-scFv attenuates the HFD-induced loss of cortical bone and increases cancellous bone. **a** Femoral BMD in 6.5-month-old male mice (LDLR-KO, ND, n = 10; LDLR-KO, HFD, n = 11; LDLR-KO;E06-scFv, HFD, n = 10). **b** Femoral cortical bone architecture in 7.5-month-old male mice (cortical thickness: LDLR-KO, ND, n = 10; LDLR-KO, HFD, n = 11; LDLR-KO;E06-scFv, HFD, n = 10; area measurements: LDLR-KO, ND, n = 10; LDLR-KO, HFD, n = 8; LDLR-KO; E06-scFv, HFD, n = 10). **c** Histomorphometry of the endocortical femoral surface (osteoblast and osteoclast numbers: LDLR-KO, ND, n = 10; LDLR-KO, HFD, n = 8; LDLR-KO;E06-scFv, HFD, n = 9; BFR/BS: LDLR-KO, ND, n = 9; LDLR-KO, HFD, n = 9; LDLR-KO;E06-scFv, HFD, n = 7). **d** Femoral cancellous bone architecture at 7.5 months (LDLR-KO, ND, n = 10; LDLR-KO, HFD, n = 11; LDLR-KO;E06-scFv, HFD, n = 10). **e** Histomorphometry of the cancellous femoral surface (osteoblast and osteoclast numbers: LDLR-KO, ND, n = 10; LDLR-KO, HFD, n = 8; LDLR-KO;E06-scFv, HFD, n = 10; BFR/BS: LDLR-KO, ND, n = 10; LDLR-KO, HFD, n = 9; LDLR-KO;E06-scFv, HFD, n = 9). Data analyzed by ANOVA except for medullary area which were analyzed by ANOVA on Ranks. Data shown represent the mean (±SD). Ob.N osteoblast number, Oc.N osteoclast number, BFR bone formation rate, BS bone surface, NS not significant

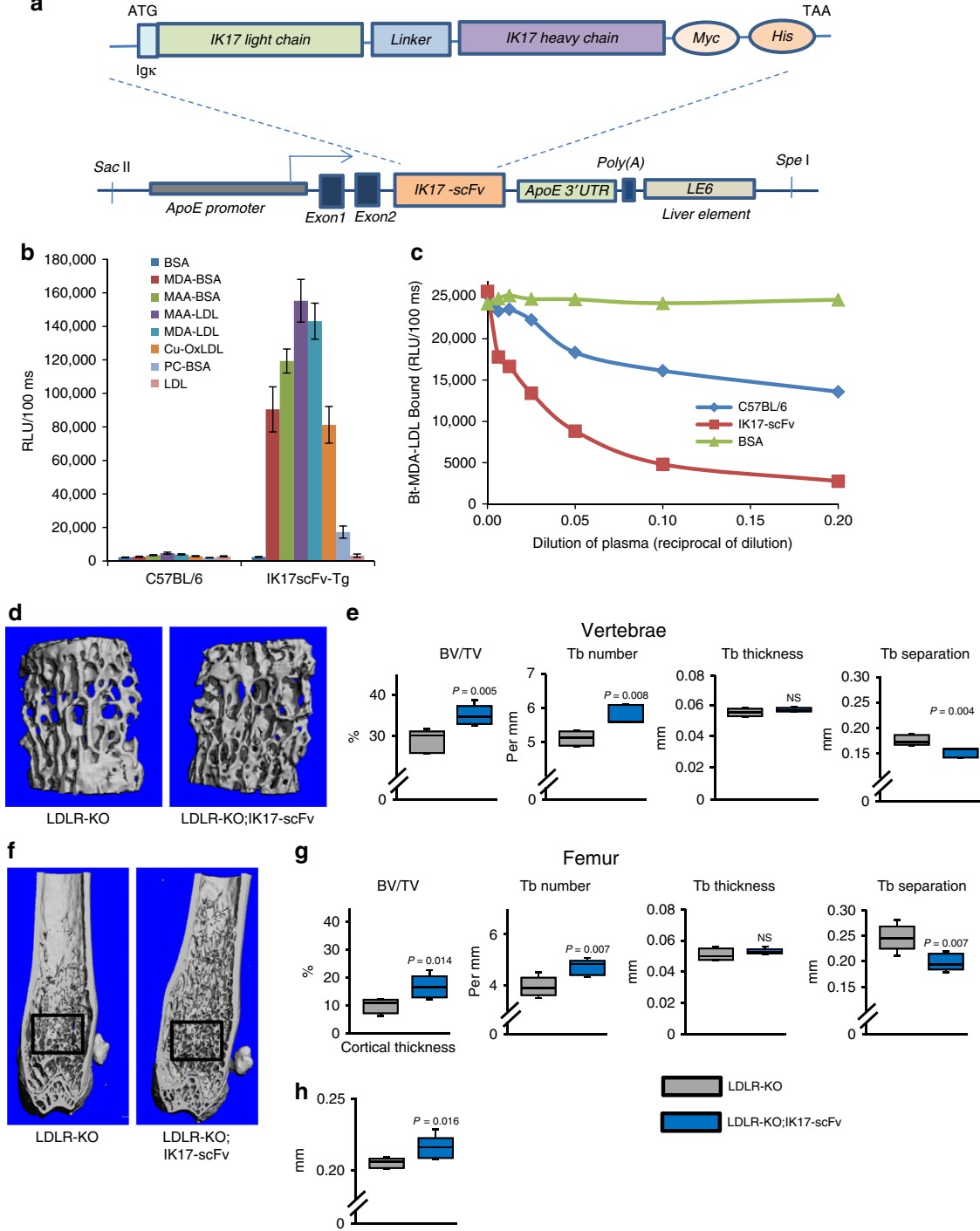

**Fig. 4** IK17-scFv increases cancellous bone and cortical thickness in HFD-fed mice. **a** IK17-scFv transgene structure. The transgene consists of a mutated single chain variant of the antigen binding domain of the human anti-MDA antibody (IK17). The transgene contains a fusion protein of the heavy- and light-chain variable domains, covalently linked by a flexible peptide linker. The transgene is expressed under the control of the ApoE promoter. **b** Binding properties of plasma from wild-type C57BL/6 and IK17-scFv-Tg mice ($n = 6$ per group) to indicated antigens. Data shown represent the mean (±SEM). BSA: bovine serum albumin, MDA-BSA: malondialdehyde-modified BSA, MAA-BSA: malondihyldehyde-acetaldehyde-modified BSA, MAA-LDL: malondihyldehyde-acetaldehyde-modified LDL, MDA-LDL: malondialdehyde-modified LDL, Cu-OxLDL: copper-oxidized LDL, PC: phosphocholine. **c** Inhibition of binding of biotinylated MDA-modified LDL (Bt-MDA-LDL) to J774 macrophages by IK17-scFv plasma. **d**–**h** Micro-CT determination of (**d**, **e**) vertebral, and (**f**–**h**) femoral bone architecture in 6-month-old male mice fed a HFD diet. **d** Lateral images of vertebral cancellous bone. **e** Quantification of vertebral cancellous bone architecture. **f** Longitudinal images of distal femur, with diaphysis at the top. Boxes indicate area of quantification of cancellous bone. **g** Quantification of cancellous bone architecture. **h** Quantification of diaphyseal cortical thickness. $n = 5$ per group. **e**–**g** data were analyzed by Student's $t$-test except for vertebral trabecular number which were analyzed by Rank Sum test. Data in **h** were analyzed by Rank Sum test. Data shown represent the mean (±SD). BV/TV: bone volume/total volume, Tb: trabecular, NS: not significant

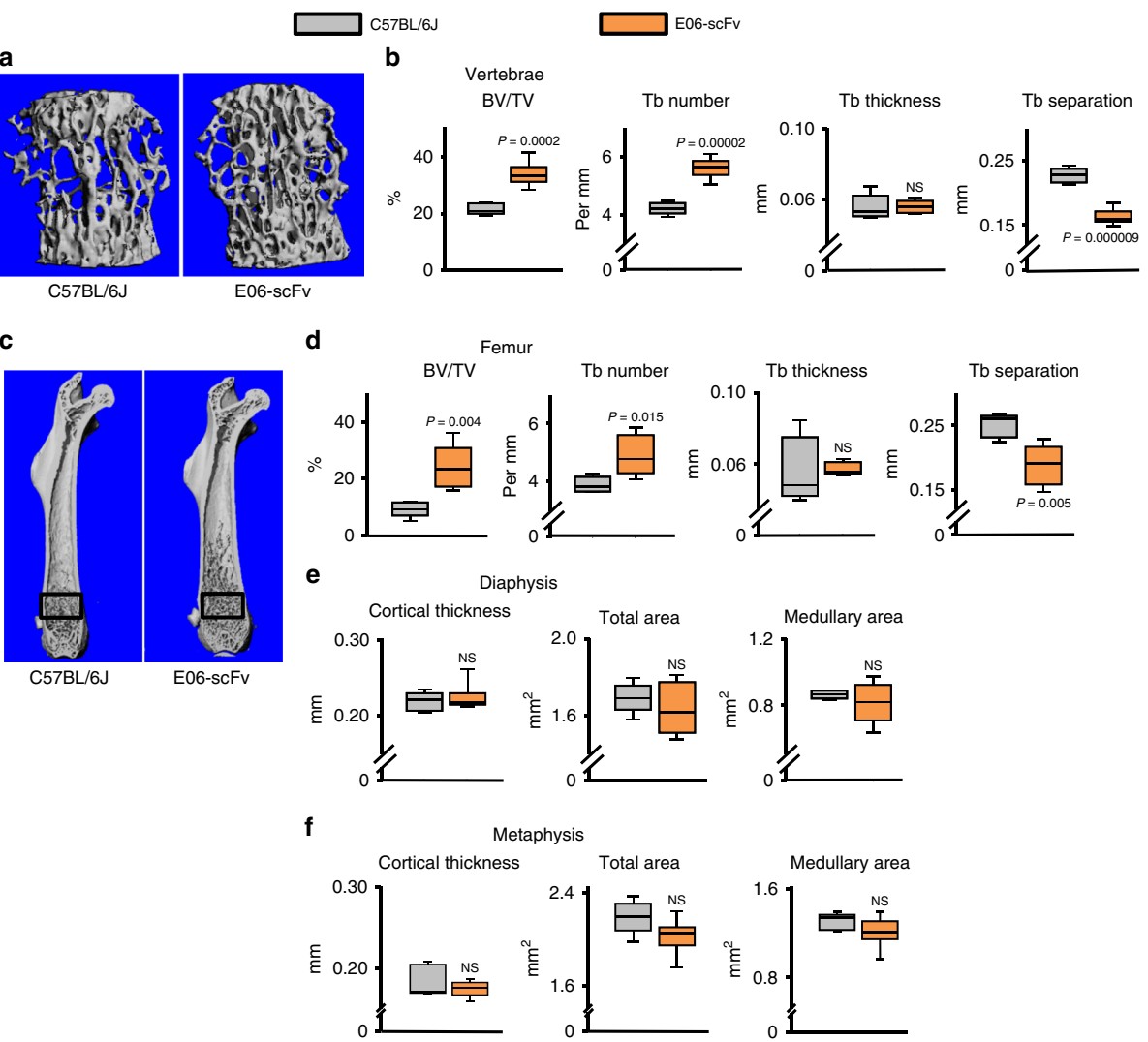

**Fig. 5** Overexpression of E06-scFv increases cancellous bone in mice fed a normal diet. Micro-CT determination of (**a**, **b**) vertebral, and (**c**–**f**) femoral bone architecture in 5-month-old male mice fed a normal diet. **a** Lateral images of vertebral cancellous bone. **b** Quantification of vertebral cancellous bone architecture. **c** Longitudinal images of femur with proximal end at the top. Boxes indicate area of quantification of cancellous bone. **d** Quantification of cancellous bone architecture. **e** Quantification of diaphyseal cortical architecture. **f** Quantification of metaphyseal cortical architecture. C57BL/6 J $n = 5$, E06-scFv $n = 6$. Data analyzed by two-tailed Student's $t$-test. Data shown represent the mean (±SD). BV/TV: bone volume /total volume, Tb: trabecular, NS: not significant

trabecular number and decrease in trabecular separation in the HFD-fed LDLR-KO mice (Fig. 3d). HFD-fed LDLR-KO;E06-scFv mice exhibited an increase in femoral cancellous bone mass as compared to LDLR-KO fed either ND or HFD, due to a substantial increase in trabecular number and decrease in trabecular separation. (Fig. 3d). HFD caused a decline in the number of cancellous osteoblasts, as well as a reduction in osteoclasts (Fig. 3e). In contrast, the number of osteoblasts was significantly higher, whereas osteoclast number was unchanged, in HFD-fed LDLR-KO;E06-scFv mice as compared to HFD-fed LDLR-KO controls (Fig. 3e). Bone formation rate (BFR) mirrored the changes in osteoblast number in the two groups, but did not reach statistical significance.

**IK17-scFv increases bone mass in HFD-fed mice**. To investigate whether the bone effects of the E06-scFv transgene were a property of anti-OSEs in general, we next examined mice over-expressing IK17-scFv, which specifically binds to the OSE MDA[24]. The transgene construct is analogous to the one used for the E06-scFv, except for different heavy chain and light-chain variable domains and linker (Fig. 4a). The IK17-scFv is present in plasma of mice at an average concentration of 40 µg/ml (range 16.2 to 64 µg/ml). IK17-scFv mice gained weight equally and did not exhibit any developmental defects. The binding specificity of the IK17-scFv transgene was determined using ELISA (Fig. 4b). Plasma from IK17-scFv transgenic mice bound to MDA or MAA (malonaldehyde-acealdehyde) epitopes but not to PC epitopes or non-oxidized LDL. Plasma from IK17-scFv transgenic mice also prevented the binding of MDA-containing LDL to macrophages compared to plasma from C57BL/6 mice (Fig. 4c).

LDLR-KO;IK17-scFv male mice and LDLR-KO controls were fed a HFD containing 60% fat from lard for 4 months, beginning at 2 months of age. HFD-fed LDLR-KO;IK17-scFv transgenic mice exhibited a significant increase in cancellous bone mass both in the vertebra (Fig. 4d, e) and in the femur (Fig. 4f, g) as compared to HFD-fed LDLR-KO controls. The increased cancellous bone mass was due to increased trabecular number and reduced trabecular separation, in both the vertebra and the femur. Moreover, LDLR-KO;IK17-scFv transgenic mice had increased cortical thickness (Fig. 4h).

**Table 1 EO6-scFv increases cancellous bone mass by increasing bone formation rate and decreasing osteoclasts**

|  | C57BL/6 J | EO6-scFv |
|---|---|---|
| Vertebral BV/TV (%) | 22.5 ± 2.2 | 29.6 ± 4.2 ($P = 0.002$) |
| Femoral metaphysis BV/TV (%) | 4.5 ± 1.6 | 14.2 ± 3.7 ($P = 0.003$) |
| Femoral diaphyseal Cortical Thickness (mm) | 0.20 ± 0.01 | 0.20 ± 0.02 (NS) |
| Vert. BFR/BS ($\mu m^2/\mu m/d$) | 0.05 ± 0.02 | 0.10 ± 0.04 ($P = 0.05$) |
| Vert. N.Oc/B.Pm (#/mm) | 8.63 ± 1.52 | 4.19 ± 1.62 ($P = 0.03$) |

Quantification of cancellous and cortical bone mass as determined by micro-CT and histomorphometric analysis of cancellous bone in 11-month-old mice fed a normal diet. $n = 4$, 2 females and 2 males per genotype. Data shown represent the mean (±SD). Data were analyzed by two-tailed Student's $t$-test
BV/TV Bone volume/total volume, BFR/BS bone formation rate/bone surface, N.Oc osteoclast number, B. Pm bone perimeter, NS not significant

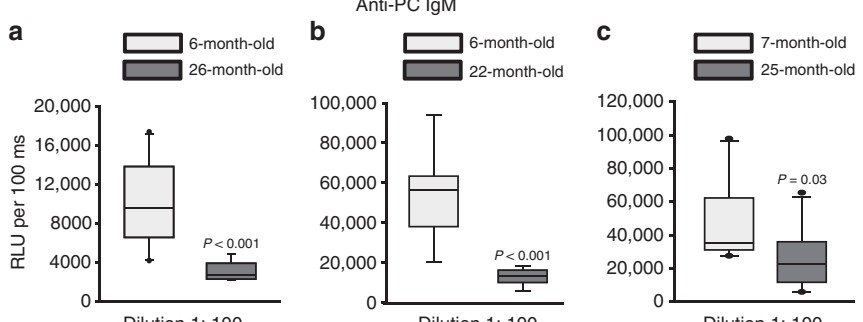

**Fig. 6** Aging is associated with reduced anti-PC IgM levels. Quantification of anti-PC IgM levels by ELISA in serum of **a** 6- ($n = 10$) and 26- ($n = 9$) month-old female C57BL/6 J mice; **b** 6- ($n = 9$) and 22- ($n = 8$) month-old female C57BL/6 J mice, and **c** 7- ($n = 10$) and 25- ($n = 10$) month-old female C57BL/6 J mice. Data analyzed by Rank Sum test. Data shown represent the mean (±SD). RFU: Relative luminescence units

**EO6-scFv increases cancellous bone in mice fed a normal diet**. Having observed an anabolic effect of two different anti-OSE antibodies in the context of atherosclerosis induced by HFD, we examined whether the overexpression of one of them (EO6-scFv) had an effect on bone of mice maintained on a normal diet. At 5 months of age, EO6-scFv transgenic mice exhibited a significant increase in cancellous bone mass in vertebrae and femora, as determined by micro-CT (Fig. 5a–d). This increase was due to increased trabecular number and reduced trabecular separation, while trabecular thickness was unchanged. Notably, the cancellous bone of the femur extended into the diaphyseal region in EO6-scFv mice—well-beyond the usual metaphyseal location of this bone compartment in wild type (WT) mice (Fig. 5c). The EO6-scFv transgene had no effect on femoral cortical bone as determined by measurements of cortical thickness, total area or medullary area at the diaphysis (Fig. 5e), or distal metaphysis (Fig. 5f). The same findings were reproduced in a second experiment employing a cohort of 11-month-old male and female EO6-scFv mice fed a normal diet (Table 1). Histomorphometric analysis of vertebral cancellous bone in these mice showed increased bone formation rate and a decrease in osteoclast number (Table 1).

**Aging is associated with reduced anti-PC IgM levels**. Age-related loss of cancellous bone in mice is due to an insufficient number of osteoblasts and it is associated with increased lipid peroxidation[39–41]. In view of evidence that B-1 cells decline with age in humans[34–37], we measured the level of anti-PC IgM antibodies in aged female C57BL/6 J mice, in which we had previously documented age-dependent decline in cortical and cancellous bone[42]. We found that, compared to 6- or 7-month-old mice, sera from 22 or 26-month-old mice had much lower levels of anti-PC IgM (Fig. 6a–c).

**Discussion**

The evidence reported herein reveals that anti-OSE antibodies recognizing the PC and MDA moieties counter the adverse effects of these oxidized lipids in mice. Unexpectedly, the EO6-scFv also promoted bone anabolism not only in the context of hyperlipidemia, but also in mice maintained on a normal diet. This finding strongly suggests that OSEs generated throughout life chronically restrain bone formation. Specifically, OSEs negatively affect osteoblasts, and may increase osteoclasts, thereby causing a previously unappreciated effect on skeletal homeostasis regardless of diet.

OSEs are ubiquitous, and the innate responses that they trigger have important physiological, as well as pathophysiological consequences[11]. OSE generation occurs during oxidation of LDL and it is also a well-recognized feature of physiological cell apoptosis, a process that continuously occurs in highly regenerating tissues, such as bone marrow and bone[11,43,44]. Apoptotic bodies and microparticles shed by apoptotic cells contain abundant OSEs identical to those formed during lipid peroxidation[11,43–45]. Such OSEs mark oxidatively modified endogenous molecules, which can be then recognized and removed by the innate immune system[11]. Indeed, these OSEs represent a class of many 'eat-me' signals recognized by macrophages during efferocytosis[46]. The only known activity of the anti-OSE scFv antibody fragments is their ability to bind to their target antigens, thereby leading to sequestration or masking of exposed PC or MDA neo-epitopes. In turn, this could either inhibit the uptake of these OSEs, and/or block their proinflammatory properties[11]. The bone anabolic effect of these antibody fragments is likely unrelated to increased clearance of apoptotic cells because this requires the Fc portion of a full antibody. Instead, their anabolic efficacy is probably due to sequestration of OSEs present on OxLDL or apoptotic cells and the inhibition of the proinflammatory and anti-osteogenic effects of such OSEs.

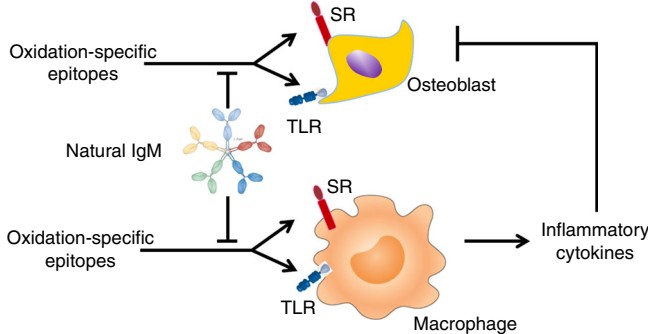

**Fig. 7** Model for the anabolic effect of anti-OSE antibodies on bone. OSEs may directly inhibit the differentiation and survival of osteoblasts via scavenger receptor (SR) or toll-like receptors (TLR). OSEs may stimulate the production of anti-osteogenic cytokines via activation of these receptors on macrophages. Natural IgM that recognize OSEs may exert an anabolic effect by preventing their binding to SR and TLR of osteoblastic cells and /or macrophages

The histomorphometric evidence presented in this paper shows that in both LDLR-KO;E06-scFv mice fed a HFD or E06-scFv mice fed a normal diet, osteoblast number and bone formation rate in cancellous and cortical bone was increased. This finding clearly indicates that the E06-scFv transgene affects osteoblast by blocking the anti-osteogenic effects of OSEs.

Scavenger receptor class B type I (SR-B1) and CD36 are receptors for OxLDL[47,48] and the two major SRs for PC-OxPL[11]. Both these receptors are expressed on macrophages, as well as osteoblasts. In osteoblasts, they have been implicated in the uptake of OxLDL, cholesteryl ester, and estradiol[49]. The role of these SRs in bone homeostasis has been studied only in mice that lack SRs globally, but the data from those studies are conflicting. Indeed, it has been reported that although mice with global deletion of CD36 have low bone mass[50], mice with global deletion of SR-B1 maintained on a normal diet have high bone mass[27,51,52], as well as striking architectural and histomorphometric similarities with E06-scFv transgenic mice. Be that as it may, the global nature of the deletion makes it difficult to understand the role of SR-B1 specifically in osteoblasts and does not rule out the possibility that the phenotype is due to effects of OSEs in other cell types. In addition to the direct effects on osteoblasts, the OSEs may act indirectly via the stimulation of anti-osteogenic cytokine (e.g., TNFα and IL1-β) production by macrophages (Fig. 7). The dissection of direct vs. indirect mechanisms will require further studies.

We have shown previously that HFD had no effect on endosteal osteoclast number, but caused a significant decrease of osteoclasts in cancellous bone[10]. The latter effect is in agreement with studies showing that RANKL-induced differentiation of the RAW264.6 pre-osteoclastic cell line is inhibited by OxLDL[53]. However, others have reported conflicting results. Namely, OxLDL increased basal levels of RANKL in MG-63 human osteosarcoma cells[54], and hyperlipidemia promoted osteoclastogenesis[55,56]. In the present work, the E06-scFv transgene did not change osteoclast number in the LRLR-KO;E06-scFv transgenic mice fed a HFD. Albeit, a decrease in osteoclasts was seen in the older E06-scFv transgenic mice. Thus, the effects of anti-OSEs on osteoclastogenesis remain presently unclear and future studies will be needed to clarify whether the OSEs exert direct or indirect effects on osteoclastogenesis or whether anti-OSEs prevent those effects.

In addition to a pathogenic role of OSEs in the setting of hyperlipidemia, our data suggest for the first time that anti-OSE antibodies have the potential to protect the skeleton from adverse effects of OSEs; and that the titer of endogenous anti-OSE

antibodies is insufficient to maintain maximal cancellous bone mass under normal conditions. The clinical relevance of anti-OSE-specific IgM has recently been elucidated by the evidence that low levels of anti-PC IgM antibodies are associated with increased incidence of cardiovascular diseases[11]. In humans B-1 cells decline with age[34–37], raising the possibility that declining NAb levels against OSEs contribute to the pathogenesis of age-related diseases. The decline of anti-PC IgM levels in older mice may indicate that such decline contributes to the pathogenesis of involutional osteoporosis. Although further work will be required to establish a cause-effect association, we submit that increasing the titer of antibodies to OSEs may attenuate and perhaps reverse age-dependent bone loss. This could be achieved by immunization strategies to increase endogenous titers of OSE antibodies (either against PC and/or MDA), or by passive immunization with exogenously generated antibodies[23,24,57]. Such novel therapeutic approaches could be used for the treatment of osteoporosis and atherosclerosis simultaneuosly.

## Methods

**Animals**. All animal procedures were approved by the Institutional Animal Care and Use Committees of the University of Arkansas for Medical Sciences, and by the UCSD. The transgenic E06-scFv founder line was generated in the C57BL/6 J background and backcrossed to each other to generate homozygous E06-scFv lines expressing high titers of the E06-scFv in plasma. These were crossed into LDLR-KO mice on C57BL/6 J background present in the UCSD vivarium. The transgenic IK17-scFv founder line was similarly generated and backcrossed into the LDLR-KO background. Transgenic mice were maintained as homozygous breeding pairs. Contemporaneous age- and sex-matched C57BL/6 J or LDLR-KO mice were used as controls. Offspring of E06 transgenic mice were screened both for plasma E06-scFv titer by ELISA (see below), and for the transgene by PCR genotyping of tail DNA using KAPA Mouse Genotyping Kit with the upstream primer sequence 5′-TAC AAT TGA GCT GGC TAG CCA CCA TGG AG-3′ and the downstream reverse primer sequence 5′-GCT GTA CCA AGC CTC CTC CAG ACT CCA CCA G-3′ to yield a 540-bp product. Offspring of IK17 transgenic mice were screened for the transgene by PCR genotyping of the tail DNA using KAPA Mouse Genotyping Kit with the upstream primer sequence 5′-AGC GAT TAG TGG TAC TGG TCG TAG C -3′ and the downstream reverse primer sequence 5′-GTC GAC GGC GCT ATT CAG ATC CTC-3′ to yield a 295-bp product corresponding to the nucleotide sequence. LDLR-KO mice were genotyped using the following primers LDLR- 5′-AAT CCA TCT TGT TCA ATG GCC GAT C -3′ and LDLR 5′-CCA TAT GCA TCC CCA GTC TT -3′ to yield a 350-bp product in KO animals. The transgenic mice are morphologically normal and healthy.

Animals were group housed under specific pathogen-free conditions, maintained with a constant temperature of 23 °C, a 12:12 h light–dark cycle, and had access to food and water ad libitum. The animals were matched for age, body weight, and total cholesterol prior to feeding the ND (TD22/5 Rodent Diet from Harlan Laboratories that contains 5.5% fat, 0.003% cholesterol and has 17% of kcal from fat) or HFD diet. The HFD used in experiment of Fig. 3 contains 21.2% fat, 0.5% cholesterol and has 41.8% of kcal from fat (TD00457 from Harlan Laboratories). Body weight of mice fed ND vs. this HDF was indistinguishable at the time of euthanasia: LDLR-KO on ND ($n = 10$), 28.9 ± 2.2 g; LDLR-KO mice on HFD ($n = 11$), 30.7 ± 4.9 g; LDLR-KO;E06-scFv mice on HFD ($n = 10$), 31.8 ± 4.4 g; NS by Kruskal–Wallis ANOVA on Ranks. The HFD diet used in experiment in Fig. 4 (TD12492 from Research Diets) contains 34.9% fat, 24.5% lard and has 60% of kcal from fat. Mice were injected intraperitoneally with tetracycline hydrochloride (30 µg per gram body weight) 6 days and 2 days before sacrifice to permit measurement of bone formation rate. Animals were euthanized by CO2 inhalation. Genotype was re-checked after euthanasia.

**Bone imaging**. BMD of the left femur was determined by dual-energy X-ray absorptiometry (DXA) of sedated mice (2% isoflurane) using a PIXIimus densitometer (GE Lunar) as described[58]. The mean coefficient of variation of BMD of a proprietary phantom (performed prior to each use) during the conduct of these studies was 0.40%. For micro-CT determination of skeletal archiecture, the fifth lumbar vertebra and the left femur were dissected, cleaned of soft tissues, placed in fixative solution, and either transferred to 100% ethanol (experiment in Fig. 3) or wrapped with saline and stored at −20 °C until analysis (experiment in Fig. 4 and Fig. 5). Bones were loaded into 12.3 mm diameter scanning tubes and imaged (micro-CT40, Scanco Medical, Bassersdorf, Switzerland). Scans were integrated into 3D voxel images (1024 × 1024 pixel matrices for each individual planar stack) and a Gaussian filter (sigma = 0.8, support = 1) was used to reduce signal noise. A threshold of 200 was applied to all scans, at medium resolution ($E = 55$ kVp, $I = 145$ µA, integration time = 200 ms). For cancellous bone determinations, the distal femur was scanned from the boundary between the growth plate and metaphysis towards the diaphysis to obtain 151 slices (12 µm/slice). The entire

vertebral body was scanned with a transverse orientation from the rostral growth plate to the caudal growth excluding any bone outside the vertebral body plate to obtain 233 slices. All cancellous measurements were made by drawing contours every 10 to 20 slices and using voxel counting for bone volume per tissue volume and sphere filling distance transformation indices, without pre-assumptions about the bone shape as a rod or plate for cancellous microarchitecture. In both vertebra and femur, the analysis was performed on contours of the cross sectional acquired images drawn to exclude the primary spongiosa and cortex. Cortical thickness, total area, and medullary area were measured at the femoral mid-diaphysis or distal metaphysis. The mean coefficient of variation of the micro-CT phantom (performed weekly) during the conduct of these studies was 1.23%.

**Histomorphometry**. Bones were fixed in 10% Millonig's formalin (Leica Microsystems Inc.), dehydrated in 100% ethanol, and then embedded in methyl methacrylate (Sigma-Aldrich) as described[59]. The analysis was done on 5 μm thick longitudinal sections. Measurements were made on both of the endosteal surfaces of a longitudinal section in a blinded fashion using Osteomeasure version XP 3.1 (Osteometrics Inc.). For measurement of static indices of osteoclast and osteoblast number, the sections were stained for tartrate-resistant acid phosphatase with toluidine blue counterstaining. Osteoblasts were identified as teams of cells (≥2) overlying osteoid. The determination of fluorescent tetracycline labeling was done using flurorese microscopy on unstained sections. Bone fomation rate (BFR) is defined as the distance between the double labels divided by the interval between the fluorochrome administrations and multiplied by the sum of the double labeled perimeter and one-half of the single-labeled perimeter. If only a single label was present, data were treated as a missing value for statistical purposes. Histomorphometric data are reported using the nomenclature recommended by the American Society for Bone and Mineral Research[60].

**Culture of osteoblastic cells**. Bone marrow cells were obtained by flushing the femoral diaphysis from 3- to 4-month-old C57BL/6 J mice with a-MEM medium (Invitrogen, Carlsbad, CA, USA). Calvaria cells were isolated from neonatal pups as described before[61] by sequential digestion with collagenase type 2 (Worthington, CLS-2, lot 47E17554B). Bone marrow stromal and calvaria cells were maintained in a-MEM containing 10% preselected FBS (Sigma-Aldrich), 1% penicillin/streptomycin/glutamine in presence of 1 mM ascorbate-2-phosphate up to 70% confluence. Oxidized LDL was obtained from Alfa Aesar, E06 IgM from AVANTI (Cat Number 330001 S lot number E06-19 and E06-17), mouse IgM from Santa Cruz Biotechnology (Catalog number sc-3881). Apoptotic cells were quantified by measuring caspase 3 activity in cell lysates as described previously[62]. Briefly, lysates (100 μg protein) were incubated with 50 μM DEVD-AFC in 50 mM HEPES (pH 7.4), 100 mM NaCl, 0.1% CHAPS, 10 mM DTT, 1 mM EDTA, and 10% glycerol, in the absence or presence of the irreversible inhibitor DEVD-CHO for 60 min. The released fluorescent AFC was measured in a microplate fluorescence reader FL500 (Bio Tek Instruments, Winooski, Vermont, USA) with excitation/emission wavelengths of 340/542 nm. Protein concentration in the lysate was measured using a Bio-Rad detergent–compatible kit (Bio-Rad).

For the alkaline phosphatase (ALP) activity measurement the cells were lysed in 100 mM glycine, 1 mM MgCl2, and 1% Triton X-100 at pH 10 using a buffer containing 2-amino-2-methylpropanol and p-nitrophenylphosphate (Sigma-Aldrich Inc). ALP activity was normalized to total protein concentration measured as described above. Proliferation was measured by BrdU incorporation with a kit from Roche Diagnostics following the manufacturer instructions. For all assays triplicate cultures were analyzed.

**ELISA**. For measurement of E06 single-chain antibody (E06-scFv) in serum or plasma, 96-well round-bottomed microplates were coated with phosphocholine-KLH (Santa Cruz Biotechnology) at 5 μg/ml (50 μl per well) in PBS overnight at 4 °C. The plates were then washed with PBS 3 times, blocked with 1% BSA in PBS (75 μl per well) for 45 min at room temperature, and washed three times with PBS. E06-scFv transgenic mouse serum or serum from C57BL/6 J controls (25 μl diluted 1:20 with 1% BSA-PBS) was added to the wells, and incubated for 90 min at room temperature. Bound E06-scFv was detected with anti-His tag antibody conjugated with alkaline phosphatase (Sigma-Aldrich Inc, Catalog number A5588, Clone HIS-1, Lot number 3085M-4836V), in Tris buffered saline (TBS) buffer containing 1% BSA (1:2000 dilution, 50 μl per well), followed by washing with PBS, a rinse with deionized water to remove phosphates, and the addition of 25 μl of 50% (1:1 in di H2O) LumiPhos 530 (Lumigen, Southfield, MI) as luminescent substrate. Chemiluminescence was measured as relative light units (RLU) over 100 ms using a plate Luminometer (DYNEX Technologies). To measure the absolute levels of the E06-scFv or IK17-scFv in plasma, a standard curve was prepared using purified E06-scFv or IK17-scFv, respectively. The purified E06-scFv and IK-17scFv were obtained from culture supernatants of transfected HEK293 cells and isolated with Ni-NTA agarose beads (Qioagen) according to manifacture's protocol.

To quantify serum levels of IgM that bind PC, 25 μL of diluted serum (in 1% BSA-PBS) was added to microplates coated with PC-KLH. Bound anti-PC IgM was quantified by chemiluminescence with anti-μ antibody conjugated with alkaline phosphatase at 1:10.000 dilution (Sigma-Aldrich Inc, Catalog number A6988, Lot number #SLBC1772V), as above.

All determinations were done in triplicate.

For the binding profile of plasma from IK17-scFv mice or control C57BL/6 mice the microplates were coated with the various antigens at 5 μg/ml (50 μl per well) in PBS overnight at 4 °C and then serum was added as above. The bound IK17-scFv was detected with anti-His antibody as described above.

**Macrophage binding assay**. To demonstrate the specificity of IK17-scFv binding to biotinylated MDA-LDL and ability to inibhit their binding to J774 macrophages, a chemoluminescent binding assay was performed as described[63]. Briefly, biotinylated MDA-LDL (5 μg/ml) was incubated in the absence or presence of IK17-scFv transgenic plasma or controls at various diluitions overnight at 4 °C. The supernatatans were then added to macrophages plated in 96-well microplates and the binding of biotinylated MDA-LDL was detected by AP-labeledNeutrAvidin and chemoluminescent ELISA.

**Statistics**. Data are shown as mean ±SD, or SEM where indicated; or as box plots which represent the 25th and 75th percentile of data points. Whiskers represent the 10th and 90th percentile of data points, dots represent data points outside the 10th and 90th percentile, and the line in the box denotes the median value. Numerical data presented in Table 1 are shown as the mean and SD. Statistical analysis was performed using Sigma Plot (version 12.5). Group mean values were compared, by Student's two-tailed t-test, or by ANOVA as appropriate. If the normality or equal variance assumptions for standard parametric analysis methods were not met, the data were analyzed by the Mann-−Whitney Rank Sum test or the Kruskal–Wallis One Way ANOVA on Ranks test. When ANOVA indicated a significant effect, pairwise multiple comparisons were performed and the P values adjusted using the Holm-Sidak method. P values <0.05 were considered statistically significant. Based on our previous studies, the sample size is adequate to detect changes with sufficient power[10]. For in vitro experiments replicates were of sufficient size to provide confidence in measurements. Each figure legend includes the number of mice used in the experiments. There was no data excluded. In Fig. 3 poor section quality precluded histological analysis of some samples and the numbers of samples analyzed are specified in the figure legend. All data were collected by personnel blinded to the identity of the samples.

**Data availability**. The datasets generated during and/or analysed during the current study are available from the corresponding author upon reasonable request.

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

## Acknowledgements

We thank Michela Palmieri, Han Li, Annick DeLoose, Stu Berryhill, and Kanan Vyas for technical assistance, and Jeff Thostenson for help with the statistical analysis. This work was supported by the Biomedical Laboratory Research and Development Service of the Veterans Administration Office of Research and Development (I01 BX000514 to R.L.J.), the National Institutes of Health (P01 AG-13918 to S.C.M.), the University of Arkansas for Medical Sciences Tobacco Funds and Translational Research Institute (239 G1-50893-01; 1UL1 RR-029884 to E.A.) and the University of Arkansas for Medical Sciences Barton Endowment funding (271 G1-51451-99 to E.A.); NIH HL P01-088093 (X.Q., T.S., J.L.W.) HL 119828 (J.L.W.), HL R35-135737 and HL P01 136275 (J.L.W.). The work was also supported by NIH P20GM125503.

## Author contributions

E.A., S.W., X.Q., F.Y., S.T., J.L.W., and R.L.J. developed the concept, designed the experiments and analyzed the data. E.A., S.W., F.Y., and X.Q. perfomed the analyses. R.S.W. provided advice. E.A., R.L.J., S.M., and J.L.W. wrote the manuscript with the contribuitions from all authors, who commented on it at all stages. E.A. created the figures.

## Additional information

**Competing interests:** X.Q., S.T., and J.L.W. are co-inventors and receive royalties from patents owned by the University of California San Diego on oxidation-specific antibodies. S.T. currently has a dual appointment at UCSD and as an employee of Ionis Pharmaceuticals. J.L.W. is a consultant to Ionis Pharmaceuticals. S.C.M. is a co-founder and owns equity of Radius Health Inc. The remaining authors declare no competing interests.

