## [Peer Review File · Nature Communications]

Editorial Note: This manuscript has been previously reviewed at another journal that is not operating a transparent peer review scheme. This document only contains reviewer comments and rebuttal letters for versions considered at Nature Communications. Mentions of prior referee reports have been redacted.

REVIEWERS' COMMENTS:

Reviewer #1 (Remarks to the Author):

The authors have answered most review issues satisfactorily. The only remaining issue is regarding pair-feeding versus ad libitum access to high-fat diet. At the very least caloric intake should have been measured in treated and control groups. However, the reviewer is prepared to overlook this as metabolic parameters are not being compared.

Reviewer #3 (Remarks to the Author):

The authors addressed all of my concerns and have added additional data to support their claims.

Data presented in Figure 6 is really interesting, however without any data to support at least a correlative relation, it remains speculative.

I have no more comments.

Response to reviewers

We wish to thank the editor for considering this paper for publication in *Nature Communications* and the reviewers for revising the newer version. Responses to the reviewers' concerns, in line with the editor's suggestion, are provided below. We believe that the revisions made in response to the reviewers' criticisms have considerably improved the paper and for that we are again grateful. The modified portions of the text have been marked and can be viewed using the "track changes" function in Word.

Concerns of Reviewer #1

The authors have answered most review issues satisfactorily. The only remaining issue is regarding pair-feeding versus ad libitum access to high-fat diet. At the very least caloric intake should have been measured in treated and control groups. However, the reviewer is prepared to overlook this as metabolic parameters are not being compared.

We have not measured the caloric intake in the experiment reported in Figure 3. The mice were fed ad libitum normal diet or high fat diet. However, there was no significant difference in the weight of the mice at the time of euthanasia (at 7.5 months of age). We now indicate that in the revised text (lines 268-269 on page 13). For the perusal of the reviewer, the data are shown in the table below. Therefore, it is unlikely that the difference in caloric intake could have accounted for the difference in the bone phenotype on these mice.

Weight of the mice at the moment of euthanasia (7.5 months of age)	
LDLR-KO mice on normal diet (n=10)	28.9 ±2.2 grams
LDLR-KO mice on HFD (n=11)	30.7 ± 4.9 grams
LDLR-KO;E06-scFv mice on HFD (n=10)	31.8 ± 4.4 grams

These data were analyzed with Kruskal-Wallis One Way analysis of Variance on Ranks.

Concerns of Reviewer #3

The authors addressed all of my concerns and have added additional data to support their claims. Data presented in Figure 6 is really interesting, however without any data to support at least a correlative relation, it remains speculative. I have no more comments.

We agree with the reviewer that a cause –effect association between the decline of the Anti-PC IgM in old mice and the age-related bone loss will require further studies. Therefore, we have modified the discussion (see lines 237 and 238 on page 12) .